# Physiological and Transcriptomic Analyses Reveal the Effects of Carbon-Ion Beam on *Taraxacum kok-saghyz* Rodin Adventitious Buds

**DOI:** 10.3390/ijms24119287

**Published:** 2023-05-26

**Authors:** Xia Chen, Yan Du, Shanwei Luo, Ying Qu, Wenjie Jin, Shizhong Liu, Zhuanzi Wang, Xiao Liu, Zhuo Feng, Bi Qin, Libin Zhou

**Affiliations:** 1Biophysics Group, Biomedical Center, Institute of Modern Physics, Chinese Academy of Sciences, Lanzhou 730000, China; 2University of Chinese Academy of Sciences, Beijing 100049, China; 3Guangdong Key Laboratory for New Technology Research of Vegetables, Vegetable Research Institute, Guangdong Academy of Agricultural Sciences, Guangzhou 510640, China; 4Rubber Research Institute, Chinese Academy of Tropical Agricultural Science, Haikou 571101, China

**Keywords:** *Taraxacum kok-saghyz*, adventitious buds, carbon-ion beam, physiological response, transcriptome

## Abstract

*Taraxacum kok-saghyz* Rodin (TKS) has great potential as an alternative natural-rubber (NR)-producing crop. The germplasm innovation of TKS still faces great challenges due to its self-incompatibility. Carbon-ion beam (CIB) irradiation is a powerful and non-species-specific physical method for mutation creation. Thus far, the CIB has not been utilized in TKS. To better inform future mutation breeding for TKS by the CIB and provide a basis for dose-selection, adventitious buds, which not only can avoid high levels of heterozygosity, but also further improve breeding efficiency, were irradiated here, and the dynamic changes of the growth and physiologic parameters, as well as gene expression pattern were profiled, comprehensively. The results showed that the CIB (5–40 Gy) caused significant biological effects on TKS, exhibiting inhibitory effects on the fresh weight and the number of regenerated buds and roots. Then,15 Gy was chosen for further study after comprehensive consideration. CIB-15 Gy resulted in significant oxidative damages (hydroxyl radical (OH^•^) generation activity, 1,1-diphenyl-2-picrylhydrazyl (DPPH) radical-scavenging activity and malondialdehyde (MDA) content) and activated the antioxidant system (superoxide dismutase (SOD), catalase (CAT), peroxidase (POD), and ascorbate peroxidase (APX)) of TKS. Based on RNA-seq analysis, the number of differentially expressed genes (DEGs) peaked at 2 h after CIB irradiation. Gene Ontology (GO) and Kyoto Encyclopedia of Genes and Genomes (KEGG) analysis revealed that DNA-replication-/repair- (mainly up-regulated), cell-death- (mainly up-regulated), plant-hormone- (auxin and cytokinin, which are related to plant morphogenesis, were mainly down-regulated), and photosynthesis- (mainly down-regulated) related pathways were involved in the response to the CIB. Furthermore, CIB irradiation can also up-regulate the genes involved in NR metabolism, which provides an alternative strategy to elevate the NR production in TKS in the future. These findings are helpful to understand the radiation response mechanism and further guide the future mutation breeding for TKS by the CIB.

## 1. Introduction

Natural rubber (NR) is an irreplaceable high-molecular-weight biopolymer with high economic value, which is widely used in transportation, medicine, and national defense [1]. Thus far, the rubber tree *Hevea brasiliensis* (HB) is still the traditional single source for NR [2], which results in very narrow biologic and geographic diversity. With a rapidly increasing demand, it is crucial to develop alternative sources of NR [3]. *Taraxacum kok-saghyz* Rodin (TKS), as one of the most-promising alternative rubber crops, has aroused significant interest due to its NR (the rubber content ranges between 5 and 24%) [4]. In addition, TKS contains other valuable compounds, such as storage carbohydrates, pentacyclic triterpenes, and inulin [5]. The TKS is a perennial plant from the composite family, originating from Kazakhstan and Xinjiang, China [6], which is widely distributed in low- or high-latitude climates [7]. However, TKS is still poorly competitive with HB, due to its slow growth, small biomass, and weak weed-crop competition. Therefore, the innovation of the TKS germplasm is of great value for improving its commercial competition.

The heavy-ion beam (HIB) is a type of ionizing radiation (IR) that is produced by heavy-ion accelerators, such as synchrotrons and cyclotrons [8]. Due to its unique physical and biological advantages, it has been widely used in the mutation breeding of plants. The HIB’s most-crucial physical character is the higher linear energy transfer (LET) compared to electrons, γ-, and X-rays. HIB irradiation can induce severe DNA damage, including DNA double-strand breaks (DSBs) and clustered damage [9]. Therefore, the HIB has a higher relative biological effectiveness (RBE) than γ- and X-rays [10]. It exhibits a high mutation rate and broad spectrum in breeding [11]. For instance, Okamura et al. declared that the carbon-ion beam (CIB), the most-widely used HIB, induced a wider mutation than γ- and X-rays in carnation [12]. Shikazono et al. reported that the mutation rate of the CIB was 17-times higher than that of electrons in *Arabidopsis thaliana* [13]. Ion beams were applied for mutation breeding in the 1990s in Japan [10]. As an effective mutagen, ion beams have been widely applied in plant breeding [14]. Significant progress has been made in many plants, such as ornamental plants [15,16], crops [17], trees [18], model plants [19], and economic crops [20]. Unrestricted to species with a mature genetic transformation system and high-quality genome information, CIB irradiation can induce mutations in diverse plant species by multiform sample status [21]. However, so far, the CIB has not been utilized in TKS.

To date, it is still a huge challenge to develop new cultivars of TKS via traditional crossbreeding, because of its inbreeding depression and high heterozygosity due to its self-incompatibility [6]. It is difficult to obtain stable lines of high-quality through traditional crossbreeding or mutation breeding with seeds as the starting materials. The plant tissue culture technique is a widely used method for fundamental and applied research including plant breeding and improvement, functional genomics studies, economic plant micropropagation, and the production of bioactive components from plants [22,23]. In addition, plant tissue culture often induces genetic and epigenetic instabilities, and it has been reported that the propagation of garlic seedlings by tissue culture can improve the breeding efficiency [24]. IR combined with plant tissue culture has been used for the development of new cultivars in numerous plants [25,26]. Therefore, it is a promising strategy to combine HIB radiation and the tissue culture technique for TKS germplasm improvement.

Understanding the relationship between the mutagenesis parameters (such as dose and sample status) and the mutagenic effect facilitates efficient breeding for plants. Although extensive studies have been performed on the mutagenic effects of the HIB on many plants, so far, the CIB has not been utilized in TKS [27,28,29]. The issue of how and to what extent tissue culture materials may respond to the CIB remains poorly understood. In this study, to better inform future mutation breeding for TKS by the CIB and provide a basis for dose-selection, adventitious buds, which can avoid high levels of heterozygosity resulting from self-incompatibility, were irradiated, and the dynamic changes of the growth and physiologic parameters, as well as the gene expression pattern were profiled comprehensively. We hope our research results can provide an overview of the radiation response mechanism of TKS and guide the future mutation breeding for TKS by the CIB.

## 2. Results

### 2.1. CIB Irradiation Inhibits the Growth of TKS Adventitious Buds

The irradiated samples were morphologically distinguishable from the control, and the browning degree of explants increased with the doses (Figure 1A). Compared with the control, the fresh weight (FW) of regenerated buds decreased by 30.33%, 52.98%, 60.01%, 65.47%, 71.64%, and 76.31% on the 15th day after CIB irradiation with 5, 10, 15, 20, 30, and 40 Gy, respectively (Figure 1B). The regenerated bud number decreased by 19.67%, 32.00%, 37.67%, 38.67%, 58.67%, and 65.00% on the 15th day after CIB irradiation with 5, 10, 15, 20, 30, and 40 Gy, respectively (Figure 1C). On the 30th day after CIB irradiation, the root number decreased by 40.56%, 62.99%, 69.53%, 87.66%, 93.83%, and 98.69% compared with the control (Figure 1D). These results indicated that CIB irradiation caused damage to the growth of the regenerated buds with an apparent dose-dependent effect. Therefore, the growth parameters of adventitious buds (FW, buds, roots) after CIB irradiation were considered comprehensively, and 15 Gy was finally chosen to perform the follow-up transcriptome sequencing study.

### 2.2. The Effect of CIB on Oxidative Damages and Antioxidant System of TKS Adventitious Buds

The hydroxyl radical (OH^•^) generation, 1,1-diphenyl-2-picrylhydrazyl (DPPH) radical-scavenging activity, and malonaldehyde (MDA) content of adventitious buds for 72 h were measured after CIB irradiation (Figure 1E–G). Overall, the above three indicators in the irradiated group showed significant changes at different time points and reached the peak at 6 h after irradiation. Compared to the control, all of them increased at 6, 24, and 48 h, then recovered to the control’s level at 72 h, except for the DPPH radical-scavenging activity. In detail, the OH^•^ generation at 6, 24, 48, and 72 h after irradiation increased by 96.78%, 69.27%, 41.33%, and 16.94%, respectively, whereas it peaked at 6 h after irradiation (Figure 1E). The DPPH radical-scavenging activity in adventitious buds irradiated by the CIB was significantly higher than that of the control. Compared to the control, the DPPH scavenging activity increased by 305.32%, 195.99%, 156.48%, and 80.32% at 6, 24, 48, and 72 h after CIB irradiation, respectively (Figure 1F). The MDA content was significantly increased by CIB irradiation. Compared to the control, the value increased by 63.96%, 44.87%, 37.60%, and 14.35% at 6, 24, 48, and 72 h after CIB treatment, respectively (Figure 1G). The results showed that CIB irradiation of adventitious buds could immediately cause a large amount of ROS (within 6 h) in cells in a short exposure time, and the lipid peroxidation was intensified. Through cell regulation, it returned to a normal level after 3 days.

To further analyze the effects of CIB irradiation on the antioxidant enzyme activities on adventitious buds, we measured the activities of superoxide dismutase (SOD), catalase (CAT), peroxidase (POD), and ascorbate peroxidase (APX). Overall, the activities of the above four enzymes were significantly increased after CIB irradiation (Figure 1H–K). In detail, the activity of SOD increased dramatically at 6 and 48 h after irradiation (Figure 1H); CAT activity increased at 6, 24, and 48 h (Figure 1I). After irradiation, the activities of SOD and CAT were the highest at 6 h, decreased with time, and were equivalent to the corresponding control at 72 h (Figure 1H,I). The activities of POD and APX responded quickly to irradiation, increased significantly at the end of the irradiation process, maintained high enzyme activity within one day after irradiation, and decreased slightly on the second and third day after irradiation, but still significantly higher than the corresponding control (Figure 1J,K). These results indicate that POD and APX are crucial in scavenging the ROSs of adventitious buds after CIB irradiation. In general, the activities of the antioxidant enzymes were time-dependent, maintained at a high level within 2 days after CIB irradiation, and gradually recovered to the normal level on the third day.

### 2.3. CIB Triggers Transcriptomic Distinctions in TKS Adventitious Buds

To explore the gene expression of adventitious buds after CIB irradiation, we performed the RNA-seq of adventitious buds collected 2, 6, 24, and 72 h after 15 Gy CIB irradiation. In total, this dataset was comprised of 24 samples with 200.85 Gb of clean reads, and each sample contained ≥6.47 Gb of data with Q30 quality scores of ≥93.18% (Appendix A). Most clean reads were mapped to the TKS genome [7], including 78.44–86.13% unique mapped reads (Appendix A).

Based on the screening criteria of FC ≥ 2 and adjusted *p* < 0.05, compared to their corresponding control, 5131, 2464, 472, and 421 differentially expressed genes (DEGs) were identified at 2, 6, 24, and 72 h, respectively (Figure 2A). Among the up-regulated DEG dataset derived from four time points, 64 were common DEGs, and 2032, 671, 48 and 34 DEGs were time-specific at 2, 6, 24, and 72 h, respectively (Figure 2B). In the down-regulated DEGs dataset, 21 genes were shared among all time points, and the number of specifically down-regulated genes was 2345, 1121, 109, and 106 at 2, 6, 24, and 72 h, respectively (Figure 2C). An overview of the expression profiles of all identified DEGs of adventitious buds after CIB irradiation at different time points is shown in the heatmap (Figure 2D).

### 2.4. Time-Series Expression Profile of DEGs Induced by CIB Irradiation

Since our RNA-seq data were collected at four time points, we performed a time-series differential expression analysis to identify genes and pathways with significant temporal expression changes during the radiation response process. These DEGs were grouped into sixteen clusters with dynamic and distinct expression patterns (Figure 3A). Here, we focused on DEGs that were only up-regulated in the CIB treatment group, while unchanged in the control. Among them, the expression pattern of the DEGs in Clusters 2 and 16 was only up-regulated at 2 h in the radiated group, while no changes appeared in the control group. In Clusters 3 and 14, the DEGs were only up-regulated at 6 h in the radiated group. Meanwhile, the DEGs detected in the above four clusters were further subjected to Gene Ontology (GO) and Kyoto Encyclopedia of Genes and Genomes (KEGG) pathway enrichment analysis (Figure 3B,C). The GO analysis indicated that these DEGs in the aforementioned four clusters were primarily related to DNA damage, senescence, autophagosome, plant hormones, and the protein metabolism process (Figure 3B). KEGG pathway enrichment analysis indicated that the DEGs were mainly related to pathways involved in DNA repair, antioxidant, pyrimidine/purine metabolism, and autophagy (Figure 3C).

### 2.5. GO and KEGG Analysis in Different Comparisons after CIB Irradiation

To determine the function of the identified DEGs between the control and CIB irradiation treatment samples of TKS adventitious buds, we performed GO and KEGG enrichment analysis. The enriched GO terms from all DEGs across different comparisons are shown in Figure 4. At 2 h, the significantly down-regulated pathways were mainly associated with the pattern specification process, water transport, growth, and development. At 6 h, the significantly down-regulated pathways were mainly related to photosynthesis, plant hormones, the protein metabolism process, and the response to stimulus. However, the significantly up-regulated pathways were primarily associated with DNA damage/repair, cell death, and cell morphogenesis. At 24 h, the significantly down-regulated pathways were mainly associated with plant hormones, cell cycle, and nucleic acid metabolism. Furthermore, the up-regulated pathways were mainly associated with DNA repair, cell death, and response to stimulus. At 72 h, the significantly down-regulated pathways were mainly present for the cell cycle, DNA replication, nucleic acid metabolism, and cytoskeleton (Figure 4A).

Further, KEGG enrichment analysis was conducted to characterize potential pathways in which the DEGs were involved. The top 25 pathways are listed (Figure 4B). At 2 h after radiation, the significantly down- and up-regulated pathways were primarily associated with fatty acid elongation and linoleic acid metabolism, respectively. At 6 h, the significantly down-regulated pathways mainly were associated with photosynthesis, glyoxylate, and dicarboxylate metabolism. However, the significantly up-regulated pathways were mainly associated with DNA replication, DNA repair, and pyrimidine metabolism. At 24 h, the significantly up-regulated pathways were those involved in DNA repair, DNA replication, and pyrimidine metabolism. At 72 h, the significantly up-regulated pathways were mainly associated with pyrimidine metabolism and glutathione/purine metabolism (Figure 4B).

### 2.6. DNA Replication-/Repair-Related Pathways Are Up-Regulated by CIB in TKS Adventitious Buds

We analyzed the expression profiles of DEGs associated with DNA replication and repair on adventitious buds after CIB irradiation (Figure 5). Notably, the expression of most DEGs associated with replication were down-regulated. The down-regulated DEGs were mainly involved in DNA replication licensing factor *MCM*, *MCM2-4*, *MCM6-7*, the DNA polymerase α catalytic subunit, DNA polymerase α subunit B, the DNA primase small subunit, and hypothetical protein LAST_2X63520. The down-regulated DEGs associated with DNA replication revealed that DNA damage caused by CIB irradiation resulted in cell cycle arrest.

At the same time, DEGs in the DNA repair pathways were detected, including homologous recombination (HR), non-homologous end-joining (NHEJ), nucleotide excision repair (NER), and base excision repair (BER) (Figure 5). The DEGs involved in the HR pathway after CIB irradiation were mainly ATP-dependent DNA helicase Q-like 1, Q-like 4A, DNA repair protein recA homolog 1, protein breast cancer susceptibility 2 homolog B-like, protein chromatin remodeling 25, DNA topoisomerase 3-beta, DSS1/SEM1-like protein, and SNF2 domain-containing protein CLASSY 1-like. In the NHEJ pathway, the CIB can induce *KU70* isoform X1, X2, *KU80*, DNA repair protein *XRCC4*, and *DNA ligase 4* (*LIG4)*. The NER pathway involves RNA polymerase II transcription factor B subunit 4, DNA excision repair protein ERCC-1 isoform X1, DNA repair protein *RAD4*, and protein damaged DNA-binding 2. Regarding the BER pathway, the poly (ADP ribose) polymerase (*PARP*) 1, *PARP* 2, hypothetical protein *LSAT_3X140620,* and uncharacterized protein LOC111918964 genes were up- or down-regulated (Figure 5). Most DNA damage repair genes were up-regulated after CIB irradiation. The DEGs at 2 and 6 h after irradiation were much more than those at 24 and 72 h, indicating that the repair time of DNA damage was mainly within one day after acute irradiation.

### 2.7. Plant Hormone signal Transduction Is Involved in TKS Adventitious Buds’ Response to CIB

Plant hormones not only participate in morphogenesis, but also play an important role in protecting plants from environmental stress factors and regulating a variety of adaptive responses. We also focused on the DEGs associated with the plant hormone signal transduction pathway (Figure 6). In the case of 2, 6, 24, and 72 h after CIB irradiation, there were 15 DEGs in the auxin signal (auxin transporter-like protein 2, transport inhibitor response 1-like protein Os04g0395600, hypothetical protein LSAT_4X56881, auxin-induced protein 22D-like, *IAA27-30*, *auxin response factor 9-like isoform X2*, *GH3.1*, *GH3.5*, *GH3.6*, *ARG7*, *SAUR23*, *SAUR36*, *SAUR50*, *SAUR71*, auxin-induced protein 6B); 3 (*histidine kinase 4-like*, *ARR5*, *ARR17*) cytokinine (CK)-mediated signal pathways; 2 (della protein GAI1-like, della protein GAI-like) in the gibberellin (GA)-mediated signal pathways; 5 (*abscisic acid receptor PYL4*, protein phosphatase 2C (PP2C), *SRK2A*, *SRK2I*, abscisic acid-insensitive 5-like protein 2) in the abscisic acid (ABA)-mediated signal pathways; 3 (EIN3-binding F-box protein 1-like, EIN3-like family protein, ethylene-responsive transcription factor 1B-like) in the ethylene (ET)-mediated signal pathway; 4 (brassinosteroid insensitive 1-associated receptor kinase 1-like, probable serine/threonine-protein kinase At4g35230, protein brassinazole-resistant 1-like, *cyclin-D3-2-like*) in the brassinosteroid (BR)-mediated signal pathways; 3 (*jasmonic acid-amido synthetase JAR1-like isoform X2*, protein TIFY 10A-like, transcription factor MYC2-like) in the jasmonic-acid (JA)-mediated signal pathway; 2 (regulatory protein NPR5, transcription factor TGA2.2-like) in salicylic acid (SA) (Figure 6). Overall, CIB irradiation resulted in the differential expression of a series of genes that are involved in multiple plant hormone signal transduction pathways. In brief, auxin, CK, GA, and BR signal pathways were negatively regulated at four time points after CIB irradiation, whereas ABA and SA had dual effects on adventitious buds after CIB irradiation. The ET and JA signal pathway was positively regulated on adventitious buds after CIB irradiation. In conclusion, multiple plant hormones displayed different expression change tendencies at different time points after CIB irradiation. Our results indicate that the response to CIB irradiation is regulated by multiple hormones and interact with each other.

### 2.8. Death-Related Pathways Are Up-Regulated by CIB in TKS Adventitious Buds

Apoptosis or programmed cell death is a highly controlled cellular process essential for plant homeostasis. To gain insight into the CIB-irradiation-mediated death signal network, 42 genes related to the death response were identified from DEGs, including DNA damage, ROSs, antioxidants, autophagy, and cell death (Figure 7). Expression profiles showed that 12, 12, 4, and 2 genes were up-regulated after 2, 6, 24, and 72 h CIB irradiation treatment, respectively whereas 10, 15, 3, and 1 gene were down-regulated after 2, 6, 24, and 72 h of CIB irradiation treatment, respectively. This indicated that more death-responsive genes were activated within 6 h after CIB irradiation.

### 2.9. CIB Down-Regulates Photosynthesis and Up-Regulates Rubber-Metabolism-Related Pathways in TKS Adventitious Buds

Photosynthesis is closely related to biomass and yield in crops. CIB irradiation treatment altered the expression pattern of photosynthesis (64 DEGs) and rubber-metabolism-related genes (7 DEGs) in TKS (Figure 8). Among these, 96.88% (62/64) of the photosynthesis-related genes were significantly down-regulated at 6 h after CIB treatment, and 57.14% (4/7) of rubber-synthesis- and metabolism-related genes were significantly up-regulated at 2 h after irradiation treatment. As for DEGs related to photosynthesis, 4, 2, 1, and 0 genes were up-regulated after 2, 6, 24, and 72 h of CIB irradiation treatment, respectively; whereas 1, 62, 0, and 0 genes were down-regulated after 2, 6, 24, and 72 h of CIB irradiation, respectively (Figure 8). This indicated that the CIB treatment negatively regulated the photosynthesis in TKS. As for DEGs related to rubber synthesis and metabolism, 4 (evm.TU.utg26475.2, evm.TU.utg11760.7, evm.TU.utg21196.5, evm.TU.utg17247.12), 0, 0, and 0 genes were up-regulated after 2, 6, 24, and 72 h of CIB irradiation treatment, respectively, whereas 1 (evm.TU.utg34347.9), 3 (evm.TU.utg17247.12, evm.TU.utg24784.10, evm.TU.utg1943.6), 0, and 0 genes were down-regulated after 2, 6, 24, and 72 h of CIB irradiation, respectively. This indicated that CIB treatment can up-regulate the genes involved in rubber synthesis in TKS at 2 h after CIB.

## 3. Discussion

### 3.1. The Optimum CIB Dose Range for TKS Adventitious Buds

The dose is an important parameter for crop mutation breeding by the CIB. The number of mutations elevated with increasing dose, but the survival rate of irradiated plants decreased, which requires a huge screening population [30,31]. The inhibition effects of the CIB on adventitious buds’ growth were enhanced in a dose-dependent manner, especially for the roots, and the highest dose (40 Gy) of irradiation caused tissue browning and death (Figure 1). In our study, the synthesis and signal transduction of auxin (NAA) and cytokinin (6-BA), which are involved in root development, were mainly down-regulated (Figure 6), as well as DEGs in the photosynthesis-related pathways (Figure 8). Previous studies have shown that senescence, various modes of cell death, including apoptosis, necrosis, autophagy, and mitotic mutations, were caused by IR [32]. Similar to our study, the DEGs involved in the above pathways were mainly up-regulated by the CIB (Figure 7). According to the data of FW, regarding the number of regenerated buds and roots, the dose above 20 Gy was not suitable for TKS adventitious buds, as the roots were seriously inhibited. Taken together, 10–15 Gy of CIB irradiation is recommended for TKS adventitious buds, and samples derived from CIB-15 Gy were chosen for further study.

### 3.2. Transcriptional Map of the CIB Radiation Response in TKS Adventitious Buds

In this study, we conducted a time series analysis to classify differentially expressed genes (DEGs) based on their expression trends over time (Figure 3A). Our analysis focused on genes that were significantly induced (up-regulated) in CIB irradiation group, but were unchanged in the control group; finally, four clusters were isolated (Figure 3A). The expression level of most DEGs in Clusters 2 and 16 peaked at 2 h after CIB irradiation, while those of Clusters 3 and 14 peaked at 6 h. GO and KEGG analysis revealed that the genes in Cluster 2/16 were mainly enriched in the processes or pathways of protein modification and non-enzymatic antioxidant, and the genes in Cluster 3/14 were primarily involved in autophagy, senescence, and damage repair (Figure 3B,C). In addition to those genes that were upregulated by radiation, we also performed GO and KEGG enrichment analysis for all the DEGs, including both up-regulated and down-regulated genes. Our results showed that there were more pathways for DEG enrichment after 6 h of CIB irradiation, with more DEGs related to damage repair in these pathways (Figure 4A,B). This result is consistent with the findings of our temporal analysis (Figure 3B,C), indicating that DNA repair was performed on TKS adventitious buds 6 h after CIB irradiation. In summary, these results indicate that radiation first caused oxidative stress on TKS and rapidly activated the antioxidant system, then the DNA damage repair response was activated, and finally, the autophagy, senescence, or death processes were initiated according to the degree of damage [32,33]. This information can provide more direct and clear hints for a better understanding of the process of CIB radiation on TKS at the molecular level.

### 3.3. CIB-Induced Oxidative Damage Response at Physiology and mRNA Levels in TKS Adventitious Buds

IR damages cellular components either directly (ionization of atoms or molecules in a target) or indirectly (generating free radicals due to the radiolysis of water molecules), leading to toxic effects and excessive accumulation of ROSs, such as O^2-^, OH^•^, ^1^O_2_, and H_2_O_2_ [34]. These will further trigger the oxidative damage to lipids, proteins, and DNA, ultimately resulting in cell death [35,36]. Wang et al. showed that CIB irradiation resulted in an increase of the ROSs in *Arabidopsis thaliana* seedlings [37]. The reaction of excess ROSs and polyunsaturated fatty acids in cell and organelle membranes is the main cause of cell membrane damage [38], and MDA is a marker of lipid peroxidation. In this study, both OH^•^ generation activity, DPPH radical-scavenging activity, and MDA content in adventitious buds were significantly higher than the control within 48 h after CIB irradiation (Figure 1E–G). In addition, the transcriptome data showed that the expression patterns of genes related to ROS synthesis and metabolism changed significantly after CIB irradiation compared with the control. At 6 h after CIB irradiation, all DEGs related to ROS synthesis and metabolism were significantly down-regulated. Only one gene was up-regulated (evm.TU.utg9538.5) or down-regulated (oxidoreductase NAD-binding domain) at 24 and 72 h after CIB irradiation (Figure 8). In conclusion, the expression of ROS synthesis and metabolism genes was the most active at 6 h after CIB irradiation.

Plants possess a complex antioxidant system for ROS detoxification, which is mainly achieved by either an ROS-scavenging enzyme, such as SOD, CAT, POD, and APX, or non-enzymatic antioxidants such AsA and GSH [39,40,41]. In this study, the activities of the SOD, CAT, POD, and APX of adventitious buds were significantly increased after CIB irradiation. SOD and CAT activities peaked at 6 h after CIB irradiation (Figure 1H,I), whereas POD and APX activities showed an increasing trend and peaked at 24 h (Figure 1J,K). In *Arabidopsis* seedlings, SOD, CAT, POD, AsA, and GSH activities increased after CIB irradiation, differing from our results, as these activities peaked at 12 h [42]. Consistent with the physiological results, the transcription levels of genes related to antioxidant enzymes and peroxisome in adventitious buds also significantly changed after CIB irradiation (Figure 7). In the case of wheat irradiated by CIB and γ-rays, the proteomic data also showed significant changes in the genes or proteins of POD, CAT, APX, glutamate decarboxylase, delta-1-pyrroline-5-carboxylic acid synthase, and succinate semialdehyde dehydrogenase, in 5-day-old seedlings [43]. Taken together, the antioxidant system works synergistically to scavenge excess ROS and maintain normal growth under radiation damage stress.

### 3.4. Synergistic Effects of Multiple DNA Repair Pathways Are Induced by CIB in TKS Adventitious Buds

Most genetic mutations likely arise from errors of DNA replication and DNA damage repair [44,45]. CIB is a high LET radiation that is densely ionizing, and it always causes complex DNA damages (clustered damages), which are not only difficult to repair, but may also require the synergistic effects of multiple DNA repair pathways [46,47]. This can be proven by our transcriptome data, for instance DEGs involving HR, NHEJ, NER, BER, RE, and MMR were detected in samples irradiated by CIB (Figure 5). Among those repair pathways, NHEJ, single-strand annealing, etc., are reported to be error-prone [48,49]. In fact, for DSB damage, HR and NHEJ are two major mechanistically different repair pathways. HR is a high-fidelity repair pathway that occurs mainly during the S and G2 phases of the cell cycle and often viewed as a high-fidelity pathway [50,51]. However, some studies also indicated that HR caused structural variation such as deletions and duplications [52,53]. NHEJ repair is an error-prone repair pathway that does not depend on the homologous template, and its repair process is not restricted to a certain stage of the cell cycle, often introducing small insertions and deletions in the genome [54,55]. The *KU70*/*KU80* heterodimer and *LIG4* are two important factors in the NHEJ repair pathway [56,57]. In this study, the transcriptome data showed that *KU70*- (evm.TU.utg10215.20, newGene_10228), *KU80*- (evm.TU.utg4232.2, evm.TU.utg31120.2), and *LIG4*- (evm.TU.utg4417.10) related genes were significantly up-regulated within 72 h after CIB irradiation (Figure 5). For the HR pathway, genes related to *RECQL1*, *RECQL4*, *RECA1*, *BRCA2*, *TOP3B*, *DSS1/SEM1*, and *SNF2* were mainly up-regulated by CIB irradiation. The damage repair still occurred at 72 h after CIB irradiation, indicating that the damage caused by CIB irradiation lasted as long as 72 h, and may even last longer. This result is consistent with the physiology results where ROS (Figure 1E–G) and antioxidant enzyme activities (Figure 1H–K) remained higher than the control at 72 h.

## 4. Materials and Methods

### 4.1. Plant Materials and Growth Conditions

The seeds of TKS were obtained from TKS Rubber Technology Development and Innovation Alliance. The plants were grown under natural light conditions in a greenhouse. The regeneration procedure of the leaf is described briefly below. Firstly, tender leaves of TKS were soaked in 75% ethanol for 30 s and subsequently immersed in ten-times-diluted commercially available sodium hypochlorite solution for 8–10 min. Secondly, ten sterile 0.7 × 0.7 cm explants were placed in 90 mm Petri dishes with an induction medium, which consisted of Murashige & Skoog (MS) medium complemented with 1.2 mg/L 6-benzyladenine (6-BA, Sigma, St Louis, MO, USA), 0.05 mg/L α-naphthaleneacetic acid (NAA, Sigma, St. Louis, MO, USA), 2% sucrose (China National Pharmaceutical Group Co. Ltd., Shanghai, China), and 0.65% agar (Beijing Solarbio Science & Technology Co. Ltd., Beijing, China). Explants were incubated in the culture rooms with 24 ± 2 °C under 4000 Lux light with 16 h light and 8 h darkness. One month later, adequate adventitious buds of TKS for CIB irradiation were obtained.

### 4.2. Irradiation Treatment

The initial energy of the CIB provided by the Heavy Ion Research Facility in Lanzhou (HIRFL) at the Institute of Modern Physics, Chinese Academy of Sciences (IMP-CAS) was 967 MeV, and the average LET through the samples was 34 keV/μm. The irradiation doses were 0, 5, 10, 15, 20, 30, and 40 Gy, and the dose rate was 20 Gy/min. The TKS adventitious buds were transferred to φ 35 mm Petri dishes covered with 1.5 mL of 1/2 MS medium at 48 h before irradiation. Each dose contained 54 dishes with 6 adventitious buds per dish. After irradiation, the adventitious buds were transferred to a square Petri dish (10 cm × 10 cm) covered with fresh 1/2 MS medium (hormone-free), a total of 12 square Petri dishes for each dose with nine adventitious buds in each dish.

### 4.3. Morphological Observations

After 15 days, the FW and the number of regenerated buds were measured. The buds were cultured for a further 15 days, then the number of roots was detected. Each experiment had three biological replicates, and each replicate contained 18 adventitious buds from two square Petri dishes.

### 4.4. Measurement of OH^•^, DPPH Radical-Scavenging Activity, and MDA Content

Sampling: At 6, 24, 48, and 72 h after irradiation, the adventitious buds of the 15 Gy CIB treatment group were sampled, and the non-irradiated group was the control group. After determining the FW, the adventitious buds were rapidly frozen in liquid nitrogen and stored at −80 °C for the biochemical parameters’ determination.

Extraction: The extraction of adventitious bud samples for the biochemical parameters was adapted from the methods in Asghar et al. [58] and Dhindsa et al. [59]. Adventitious buds (0.5 g of FW) were homogenized in 5 mL of 50 mM phosphate-buffered solution (pH = 7.8) at 4 °C and centrifuged at 10,000 r/min for 20 min.

The determination of OH^•^, DPPH, and MDA: The OH^•^ generation activity was determined by following the method described by Wu et al. [60]. DPPH radical-scavenging activity was determined as described by Zhang et al. [61]. MDA content was assayed according to the method of [56].

### 4.5. Determination of Antioxidant Enzyme Activity

We detected the activities of SOD, CAT, POD, and APX using the obtained supernatant (as shown in Section 2.4). Determination of antioxidant enzyme activity: the activity of SOD was measured by the nitro blue tetrazolium chloride illumination method at 560 nm; the activity of CAT was determined by the hydrogen peroxide decomposition method at 240 nm for optical density; the activity of POD was determined by the guaiacol method at 470 nm; the activity of APX was determined by the hydrogen peroxide decomposition method at 290 nm for optical density [62].

### 4.6. RNA Isolation and Sequencing

Total RNA from 0.1 g adventitious buds was extracted by using the RNA prep Pure Plant Kit (Polysaccharides & Polyphenolics-rich, TIANGEN, Beijing, China). The purity of the RNA was examined using the NanoPhotometer spectrophotometer (IMPLEN, Westlake Village, CA, USA). The RNA concentration was measured using the Qubit RNA Assay Kit in Qubit 2.0 Flurometer (Life Technologies, Camarillo, CA, USA). The RNA integrity was assessed by the RNA Nano 6000 Assay Kit of the Agilent Bioanalyzer 2100 system (Agilent Technologies, Santa Clara, CA, USA). Following the quality verification of the extracted RNA, 1 µg RNA was transcribed into the cDNA libraries by Transcriptor cDNA Synth. The cDNA libraries were sequenced on an Illumina high-throughput platform (Illumina HiSeq 6000 platform (150 bp PE)), which is based on sequencing by synthesis (SBS) technology. The RNA-seq was completed by the Biomarker Technologies Company (Beijing, China).

### 4.7. Bioinformatic Analysis of Transcriptome Data

Raw data in the fastq format were first processed through in-house PERL scripts. Clean data were filtered by removing reads containing adapter, ploy-N, and low-quality reads from raw data. The clean data were mapped to the TKS genome (GWH; http://bigd.big.ac.cn/gwh/, accessed on 21 December 2021) using TopHat. The transcripts or gene expression levels were further quantified by the fragments per kilobase of exon per million fragments mapped (FPKM) method [63]. DEGs were identified based on comparing gene expression levels in the irradiated and control samples. DESeq2 [64] was used for identifying DEGs, and the fold change (FC) ≥ 2 and their Bonferroni corrected *p*-value (adjusted *p*) was < 0.05 were used as the screening criteria for differential expression analysis [65]. GO annotation and KEGG enrichment for the DEGs were analyzed using the software KOBAS2.0 [66]. Pathway enrichment analysis was performed using the R package of clusterProfiler 4.0 [67]. In addition, we used the enrichment factor as the index for measuring importance.

### 4.8. Statistical Analysis

Three independent samples were used to detect RNA-Seq for each treatment. The physiological experiment was performed in three replicates. The statistical significance was examined using one-way ANOVA analysis and Student’s *t*-test. Differences between samples were considered significant when *p* < 0.05.

### 4.9. Data Availability

The datasets generated and analyzed during the current study are not publicly available due to the requirements of the scientific research projects, but are available from the corresponding author upon reasonable request.

## 5. Conclusions

In conclusion, the effects of CIB irradiation on TKS adventitious buds were revealed based on physiological and transcriptomic analyses in the present study (Figure 9). The CIB, as an effective means of physical mutagenesis, has great potential for crop improvement. Combining radiation and plant tissue culture can benefit breeding for the self-incompatible species. So far, no similar research has been reported on TKS. The radiation effects of CIB irradiation on TKS adventitious buds, for the first time, was characterized comprehensively in this study. Our results showed that CIB irradiation caused significant biological effects on the development and physiology of TKS adventitious buds. In addition, based on RNA-seq analysis, we found that the adventitious buds suffered significantly oxidative damage, and a series of DNA damage repair pathways, which are a key source for genetic mutations (Figure 5), were up-regulated. Interestingly, we found that CIB irradiation can up-regulate the genes involved in NR metabolism (Figure 8), which provides an alternative strategy to elevate the NR production in TKS in the future.

## Figures and Tables

**Figure 1 ijms-24-09287-f001:**
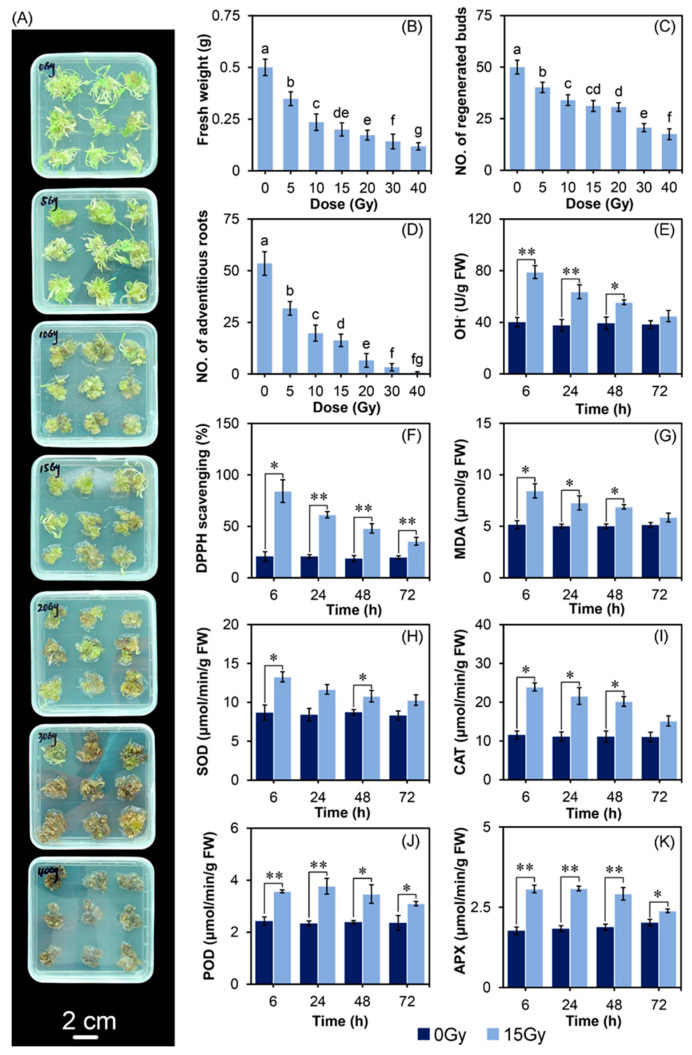
Effects of CIB radiation on development and oxidative stress physiology of TKS. (**A**) Phenotypic variation, (**B**) FW, and (**C**) the number of buds at 15 days after CIB irradiation, bar = 2 cm. (**D**) The number of roots of adventitious buds on the 30th day after CIB irradiation. (**E**) OH^•^ generation activity, (**F**) DPPH radical-scavenging activity, and (**G**) MDA content. (**H**) The activities of SOD, (**I**) of CAT, (**J**) of POD, and (**K**) of APX on adventitious buds irradiated by the CIB at doses of 0 and 15 Gy. The error bars represent the Means ± SD, data followed by the same alphabetic letters in (**B**–**D**) are not significantly difference between any two groups (*p* > 0.05) according to the Duncan’s test; * *p* < 0.05, ** *p* < 0.01.

**Figure 2 ijms-24-09287-f002:**
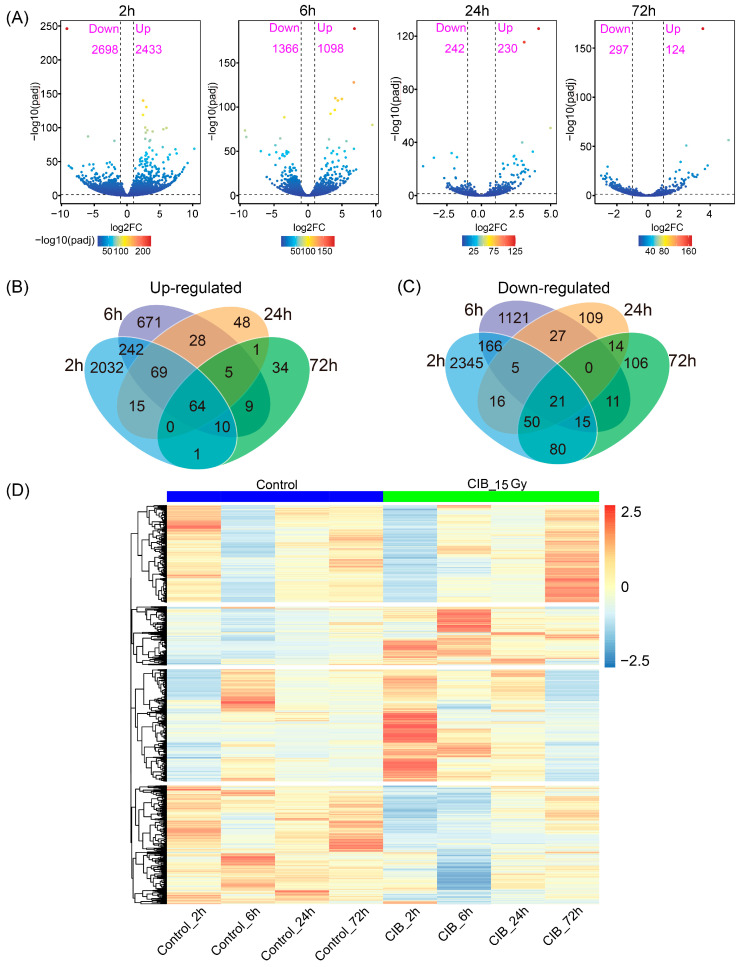
DEGs detected in TKS after 15 Gy CIB irradiation. (**A**) Numbers of up- and down-regulated DEGs in different comparisons. (**B**) Venn diagram of up-regulated DEGs in different comparisons. (**C**) Venn diagram of down-regulated DEGs in different comparisons. (**D**) Heatmap of all DEGs. Red and blue indicate up-regulated and down-regulated genes, respectively.

**Figure 3 ijms-24-09287-f003:**
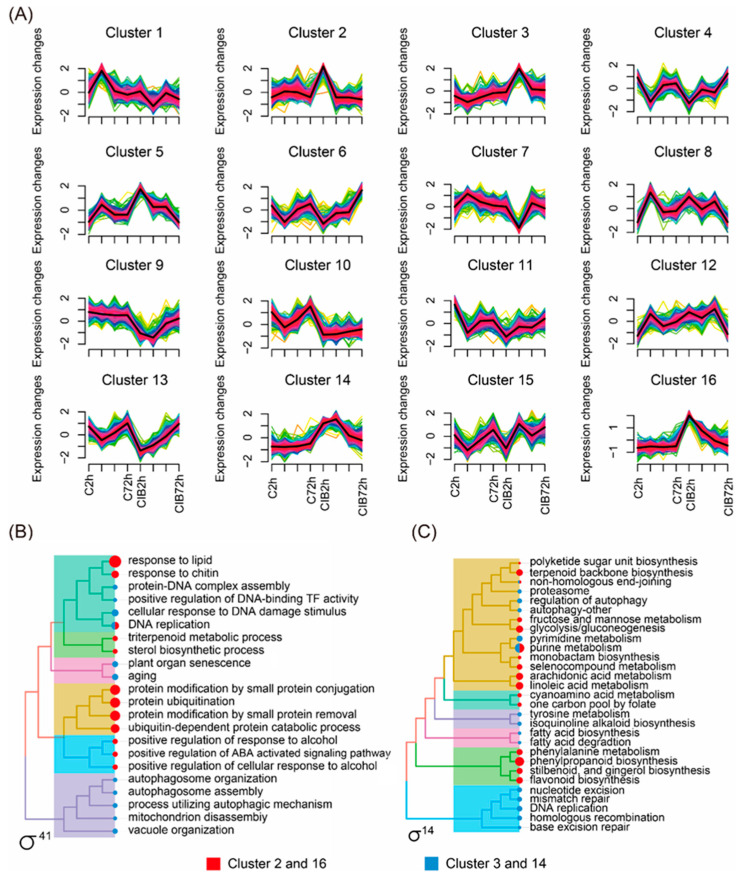
Transcriptome-wide time-series cluster of DEGs. (**A**) Cluster analysis of DEGs based on R package of ClusterProfiler 4.0. (**B**,**C**) DEGs were further categorized into different functional groups by GO and KEGG enrichment analysis. The red square represents Clusters 2 and 16, and the blue square represents Clusters 3 and 14. Yellow or green colored lines correspond to genes with low membership value; red and black colored lines correspond to genes with high membership value.

**Figure 4 ijms-24-09287-f004:**
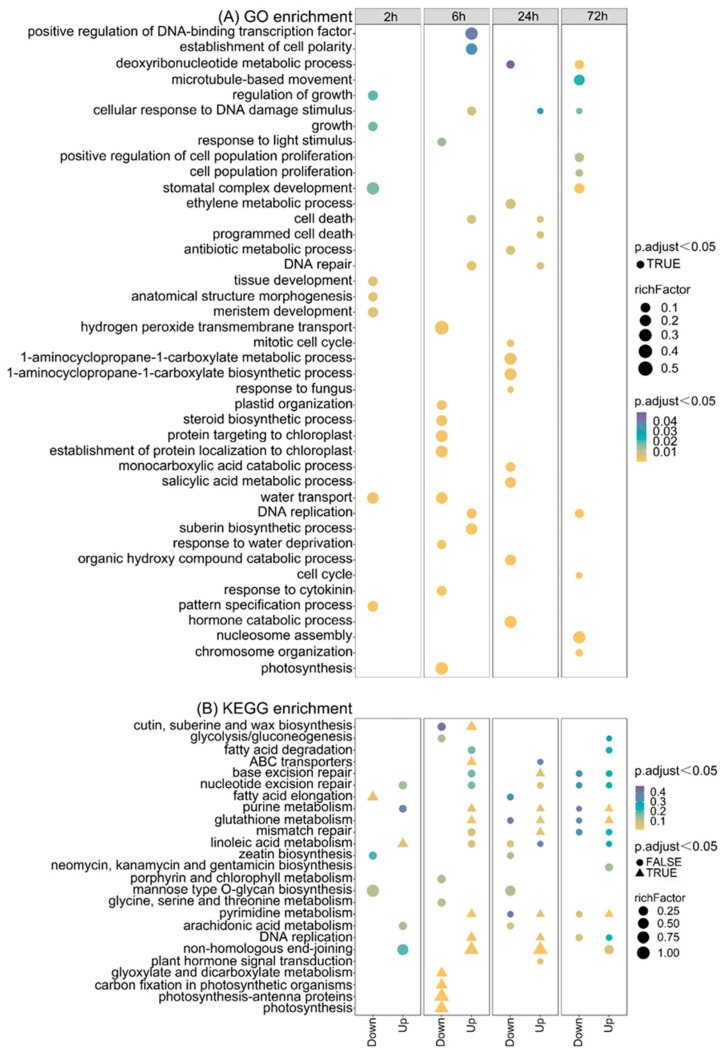
GO and KEGG enrichments of DEGs on adventitious buds after 15 Gy CIB irradiation at different time points. (**A**) GO enrichment of DEGs on adventitious buds after 15 Gy CIB irradiation. (**B**) KEGG enrichment of DEGs on adventitious buds after 15 Gy CIB irradiation. Triangles represent enriched pathways (adjusted *p* < 0.05).

**Figure 5 ijms-24-09287-f005:**
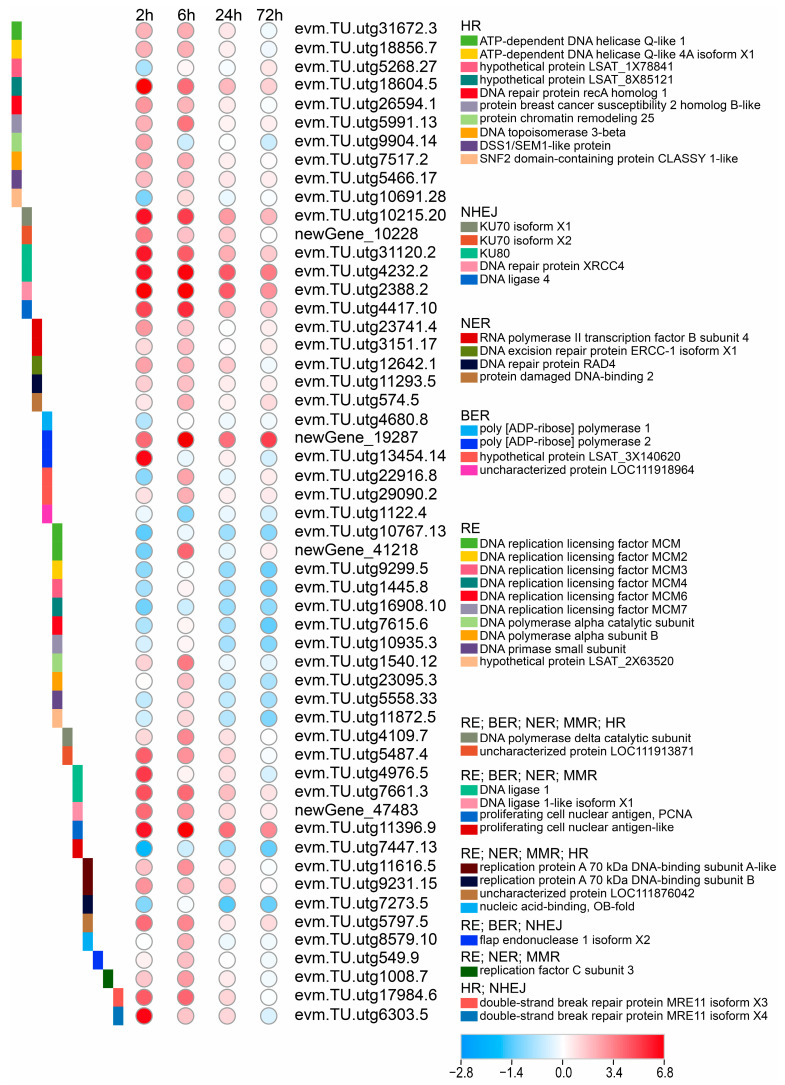
Response patterns of DNA replication/repair genes to CIB irradiation. Using log_2_ (fold change) to make the heat map, red represents up-regulated genes, and blue represents down-regulated genes.

**Figure 6 ijms-24-09287-f006:**
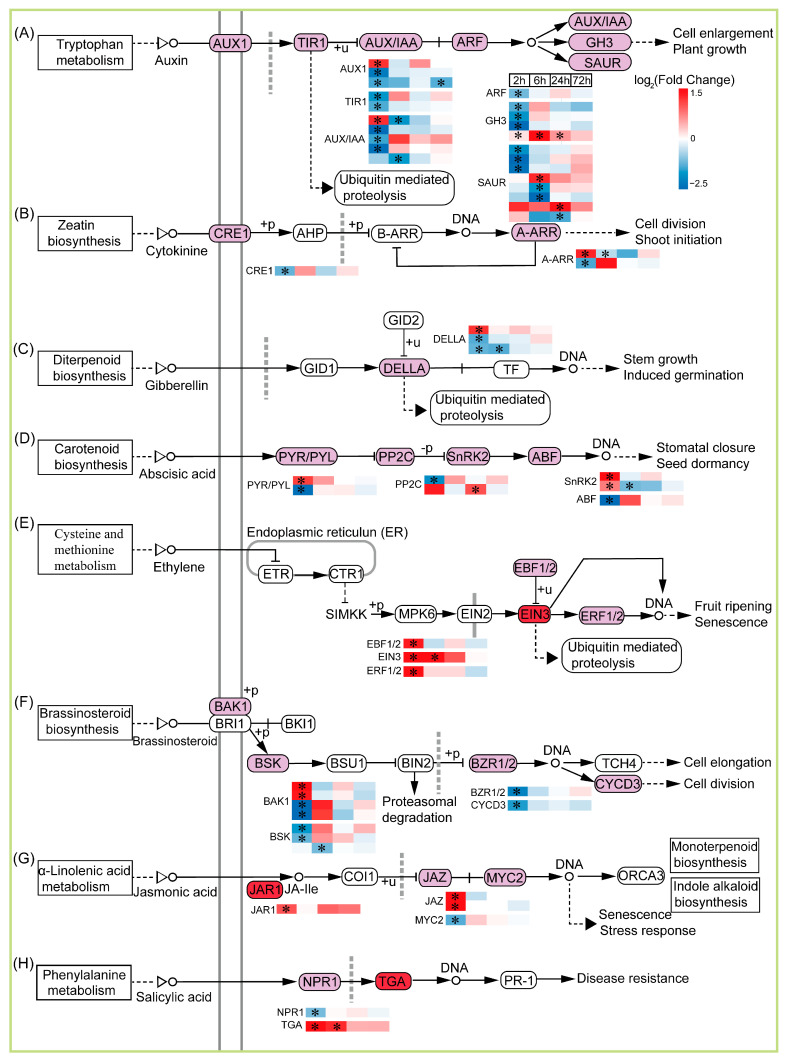
Response patterns of genes of plant hormone signal transduction to CIB irradiation. Using log_2_ (fold change) to make the heat map, red represents up-regulated genes, and blue represents down-regulated genes. “*” represents a significant difference (|log_2_ (fold change) | ≥ 1, adjusted *p* < 0.05). A–H are the changes of DEGs of different plant hormones in *Taraxacum kok-saghyz* Rodin buds treated by carbon ion beam irradiation.

**Figure 7 ijms-24-09287-f007:**
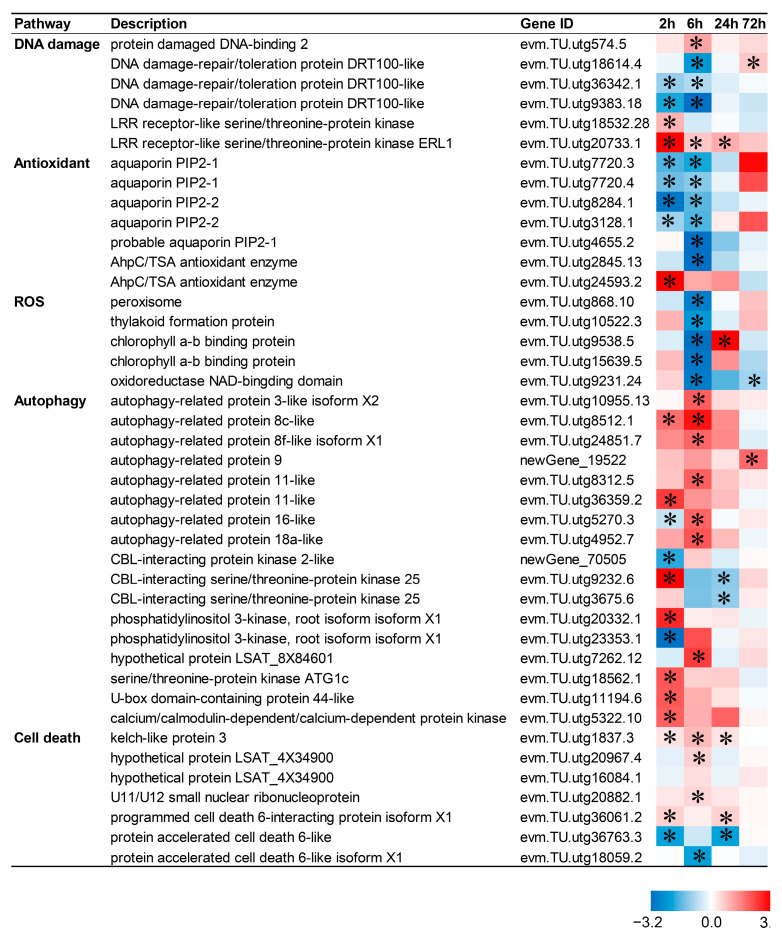
Response patterns of death-related genes to CIB irradiation. Using log_2_ (fold change) to make the heat map, red represents up-regulated genes, and blue represents down-regulated genes. “*” represents a significant difference of the gene.

**Figure 8 ijms-24-09287-f008:**
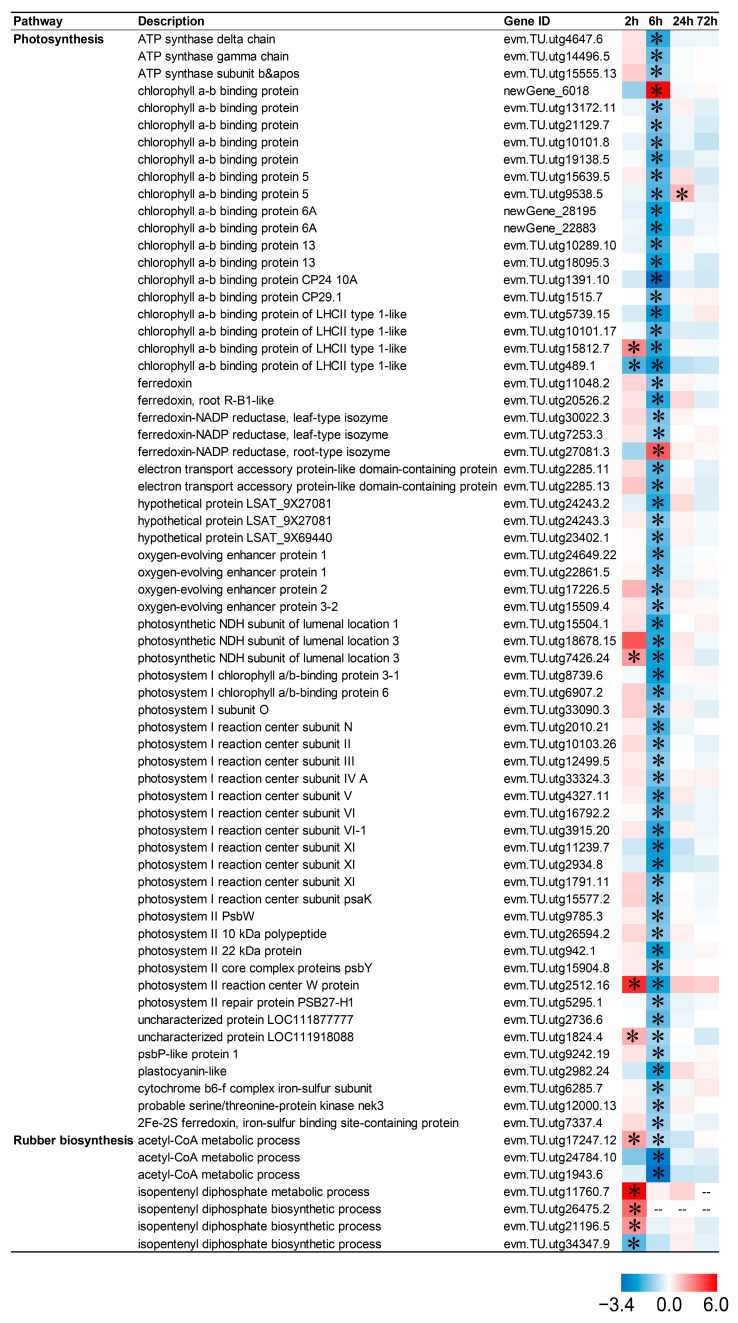
Response patterns of photosynthesis- and rubber-related genes to CIB irradiation. Using log_2_ (fold change) to make the heat map, red represents up-regulated genes, and blue represents genes down-regulated genes. “*” represents a significant difference of gene. “--" indicating that gene expression was not detected at the relevant time points after carbon ion beam irradiation.

**Figure 9 ijms-24-09287-f009:**
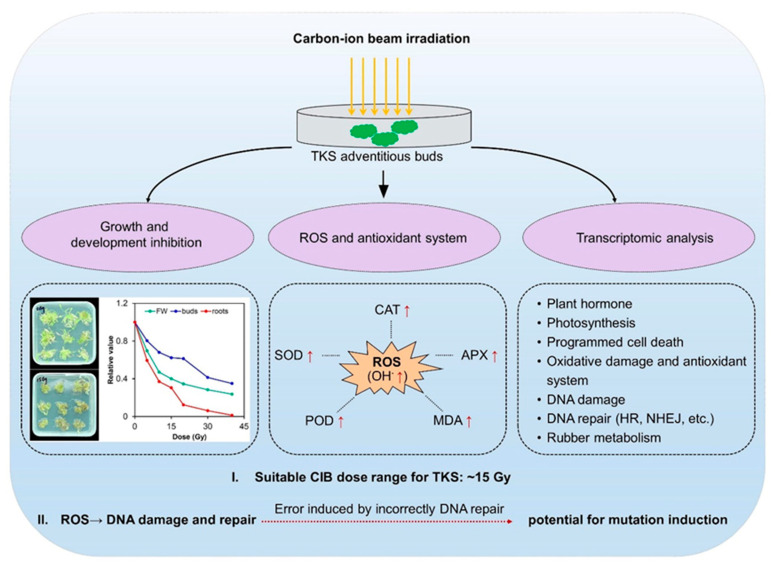
Schematic diagram of TKS adventitious bud response to CIB treatment.

## Data Availability

Data and materials will be made available upon reasonable request.

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
