# Peer review of "Physiological and Transcriptomic Analyses Reveal the Effects of Carbon-Ion Beam on Taraxacum kok-saghyz Rodin Adventitious Buds"

_ijms, 2023, doi:10.3390/ijms24119287_

Round 1
Reviewer 1 Report
After careful review, my decision is a major revision. This manuscript has good data but there are some scientific and grammatical errors and some inaccuracies observed in the manuscript. It requires improved conclusion and clarity. So, it can be accepted after exact revision. I will restrict my review to a limited number of examples rather than identifying all issues that require revision.
The main aim of the manuscript is physiological and transcriptomic analyses for elevating the NR production in TKS in the future, but the authors ignored a clear presentation of results and discussion of mutation effects on rubber content. Also, the rubber content should be considered under CIB.
Line 331: up-regulates rubber metabolism-related pathways?? Which rubber pathways are up-regulated? Please be accurate in writing. No data, results, or discussion were presented about the rubber biosynthesis pathway.
Line 483: The quality of RNA should be considered using agarose gel electrophoresis, please see the below paper for exact information
Salehi, M., Karimzadeh, G., Naghavi, M. R., Naghdi Badi, H., & Rashidi Monfared, S. (2018). Expression of key genes affecting artemisinin content in five Artemisia species. Scientific reports, 8(1), 12659.
Line 46: high-quality NR (up to 20% dry weight)??? Please pay attention that 1) The high quality of its rubber is related high molecular weight of rubber 2) the rubber content of Taraxacum kok-saghyz is ranged between 5-24%: up to 20%? Rubber content up to 20% dry weight?? Please two below papers for exact information:
Salehi, M., Cornish, K., Bahmankar, M., & Naghavi, M. R. (2021). Natural rubber-producing sources, systems, and perspectives for breeding and biotechnology studies of Taraxacum kok-saghyz. Industrial Crops and Products, 170, 113667.
Salehi, M., Bahmankar, M., Naghavi, M. R., & Cornish, K. (2022). Rubber and latex extraction processes for Taraxacum kok-saghyz. Industrial Crops and Products, 178, 114562.
Please prepare an abbreviation list
Line 24” What is Gy?
Line 37: Keywords and title should complete each other, please avoid repeated words in both
Grammatical mistakes:
Lines 22, 90, 105: add “,” before comprehensively
Line 24: 15 Gy is a new sentence, please separate it from the previous sentence
Lines 100, 101: in what respectively?
Line 107: replace this title with “The effect of CIB on oxidative damages and antioxidant system of TKS adventitious buds
Lines 108,114: Delete “activity”
Lines 116-119: Grammatical mistake, please rewrite it.
Lines 119-120: Grammatical mistake, please rewrite it.
Line 120: “Improve” is not a proper verb, please replace it
Line 369-370: of in??? Grammatical mistake, please correct it
Line 505-506: The sentence without a verb, please correct it.
Line 334-335: To better understand the action of CIB irradiation on photosynthesis and rubber process: incomplete sentence, please complete it
And other concerns
Reviewer 2 Report
This manuscript describes the response of kok-saghyz adventitious buds to irradiation at the physiology and transcriptome levels. My comments are as follows:
Fig. 1. Fresh weight, No. of buds, No. of roots are presented per a Petri dish or …?
Subsections 2.4 and 2.5 should be shortened and Fig. 3 and 4 should be moved to Suppl. Materials, since these data are not discussed in any way.
Different sampling times for physiology (6, 24, 48, and 72 h) and transcriptome (2, 6, 24, and 72 h) analyses were used. Clarify please.
The Discussion section is too short for the presented results and should be expanded.
Subsection 3.2 "CIB-induced oxidative damage response at physiology and mRNA levels in TKS adventitious buds". However, transcriptome data is described in only three lines (L.379-381) and these results are not discussed.
Subsection 3.3. In this section, almost all references are from human genetics. References from plant genetics should be cited.
Subsection 5.2. Did you use another medium after irradiation treatment (hormone-free 1/2 MS medium)?
There are no references to Supplementary Figures in the text.
Fig. S1 is identical to Fig. 1A and must be deleted.
Round 2
Reviewer 2 Report
The manuscript has been revised according to the comments and may be accepted for publication.